# Elderflowers (*Sambuci flos* L.): A Potential Source of Health-Promoting Components

**DOI:** 10.3390/foods13162560

**Published:** 2024-08-16

**Authors:** Agnieszka Nawirska-Olszańska, Joanna Kolniak-Ostek, Muhamad A. Zubaidi, Damian Maksimowski, Pavla Brandova, Maciej Oziembłowski

**Affiliations:** 1Department of Fruit, Vegetable and Plant Nutraceutical Technology, Wroclaw University of Environmental and Life Sciences, 50-375 Wrocław, Poland; agnieszka.nawirska-olszanska@upwr.edu.pl (A.N.-O.); joanna.kolniak-ostek@upwr.edu.pl (J.K.-O.); muhamad.zubaidi@upwr.edu.pl (M.A.Z.); 2Department of Functional Food Products Development, Wroclaw University of Environmental and Life Sciences, 50-375 Wrocław, Poland; damian.maksimowski@upwr.edu.pl; 3Research and Breeding Institute of Pomology, Holovousy, Ltd., 508 01 Holovousy, Czech Republic; pavla.brandova87@gmail.com

**Keywords:** elderflowers, organic acids, polyphenols, sugar, antioxidant activity

## Abstract

Elderflowers are used for both culinary and health purposes. Their composition and, therefore, their properties depend on the variety from which they were obtained. The aim of this study was to compare six cultivated varieties with the wild form. The research included determining the basic chemical composition, bioactive ingredients and antioxidant activity. This research confirms that elderberry flowers are rich in bioactive ingredients. The wild form turned out to be the most rich in bioactive ingredients, while among the tested varieties it was ‘Weihenstephan’; both of these forms were characterized by a high content of polyphenols (394.36 and 377.75 mg/100 g dm, respectively) and antioxidant activity. The PCA analysis showed general differences in the chemical composition of elderberry flowers, depending on the variety, in relation to 23 variables, as well as showing their mutual correlations and the strength of their influence on the PCA model.

## 1. Introduction

Elderberry (*Sambucus nigra* L.) is a common plant from the Adoxaceae family. Fruits and flowers are used for culinary, cosmetic and medicinal purposes. In folk medicine, elderberry flower infusions were used as anti-inflammatory, diaphoretic, expectorant, antipyretic and tonic agents. They are still used in many countries in traditional treatments for and for the prevention of colds, flu and upper respiratory tract infections. Triterpenes found in flowers increase the production and secretion of mucus in the respiratory tract and trigger the cough reflex. Elderberry flowers are believed to seal blood vessels and strengthen mucous membranes, thus reducing the risk of infection [1].

The chemical composition of elderberry flowers (*Sambuci flos*) is diverse and depends, among others, on the variety, climatic conditions and place of obtaining the raw material. The flowering time at which the harvest was conducted has a very important influence on the content of active ingredients in flowers [2]. Elderberry inflorescences should be collected in full bloom (when about 80% of the flowers are open), which, in the climatic conditions of Central Europe, takes place from the beginning of May to the end of June.

The main active substances of elderberry flowers are flavonoids, phenolic acids and their glycosides (caffeic, feluric, chlorogenic and p-coumaric acids), triterpenes (α- and β-amyrin and ursolic acid), sterols (β-sitosterol, campesterol and stigmasterol) and essential oil. They also have high antioxidant activity [2,3] so they are an important factor in the prevention of cardiovascular diseases [4].

Thanks to the content of bioactive compounds from the polyphenol group, elderberry inflorescences can be used in the prevention and treatment of atherosclerosis, because they reduce the inflammation that occurs in atherosclerotic processes, capture ROS present in the blood and prevent their formation [5]. Moreover, studies conducted on mice have confirmed that flower extracts have antidiabetic effects because they stimulate the production of insulin in the pancreas and prevent a rapid increase in blood glucose levels by slowing down its absorption [6].

The flowers also have antiseptic properties, which are mainly due to caffeic and chlorogenic acids. According to research conducted on an ethanolic extract of elderberry flowers [7], this raw material inhibits the development of five strains of bacteria, including *S. aureus*, *B. subtilis* and *P. aeruginosae*. However, in later studies conducted by Hearst et al. [8], it was shown that flowers, compared to fruits and leaves, have the highest ability to inhibit the growth of nosocomial pathogens, such as MRSA (*meticillin-resistant S. aureus*). In addition, chlorogenic acid has anti-inflammatory properties, stimulates the level of glucose in the blood and has a beneficial effect on the digestive system by stimulating intestinal peristalsis and increasing the secretion of hydrochloric acid and bile [2,4].

Elderberry flowers contain approximately 3% of triterpene compounds, which also have antioxidant and anti-inflammatory properties. It has been shown that triterpenes can reduce the number of pro-inflammatory cells because they inhibit the activity of the proteolytic enzyme elastase, which increases inflammation. Additionally, it has been confirmed that cells involved in the inflammatory process are an important element of the cancer cell environment, so it can be assumed that triterpene compounds are important in preventing the development of cancer [9].

Triterpenes also play a potential role in preventing diabetes complications because they can reduce the activity of enzymes involved in the development of diabetic kidney disease. Studies conducted on rats have shown that triterpene compounds help lower glucose levels, as well as reduce LDL and total cholesterol levels. Oleanolic acid reduces the content of sorbitol in the body, an excess of which may lead to the improper functioning of the nervous system, eyes and kidneys in diabetics [10].

In addition to the ingredients mentioned, other active substances in flowers also include organic acids, tannins, mucus, vitamins A and C and sugars. Of all the sugars found in elderberry flowers, glucose and fructose constitute the vast majority at as much as 60–85% [11]. The raw material also contains minerals (8–9%); in this regard, potassium dominates in terms of content [12].

Elderberry flowers also contain essential oils, which constitute 0.03–0.14% of the composition and consist of about 58 compounds. Together with valeric acid, they are responsible for the characteristic aroma of flower petals [13]. However, the white-cream color of the raw material and its tart taste are generated primarily by tannins and polyphenolic compounds [2].

Due to their health-promoting properties, elderberry flowers (*Sambuci flos*) are widely used as a pharmaceutical raw material, but they are becoming more and more popular in food and cosmetics industries [14]. Although the food industry is mainly interested in anthocyanins contained in elderberries, it also appreciates the properties of flowers. Importantly, they have been recognized by the Food and Drug Administration (FDA) as safe for use as a flavor ingredient. They can be used in the form of extracts that are additions to non-alcoholic drinks or can be used to produce ice cream, wine, cakes, cookies and yogurt [15].

Since the common elderberry is an important species in processing and medicine, the quality parameters of flowers from different varieties of elderberry were compared. At the same time, due to the use of mainly the wild form (variety) in processing, an attempt was made to compare the bioactive properties of elderberry flowers of cultivated varieties with the wild form. The aim of this study was to assess the content of bioactive compounds with a potentially high impact on human health in elderberry flowers of six cultivated varieties and one wild form. The research hypothesis was that the chemical composition of elderberry flowers varies in a statistically significant way depending on the variety.

## 2. Materials and Methods

### 2.1. Reagents and Standards

All standards of polyphenolic compounds and carotenoids were purchased from Extrasynthese (Lyon, France) at the purity of all reagents required for UPLC-MS. Acetonitrile, formic acid and methanol for ultraperformance liquid chromatography (UPLC; Gradient grade) were from Merck (Darmstadt, Germany). The rest of the reagents were purchased from Sigma-Aldrich (Steinheim, Germany).

### 2.2. Material

The research material consisted of flowers from various varieties of elderberry (*Sambucus nigra* L.), which were collected in May/June in 2022 in the Czech Republic (Research and Breeding Institute in Holovousy) and from one wild variety from the Kłodzko Valley (Poland) at the Śnieżnik Landscape Park. The investigations were conducted on fresh flowers. The analyses were performed on the following varieties:Albida;Bohatka;Haschberg;Sambo;Samdal;Weihenstephan;Wild elderberry.

‘Albida’, ‘Bohatka’ and ‘Sambo’ are Slovak varieties, ‘Haschberg’ is an Austrian variety, ‘Samdal’ comes from Denmark and ‘Weihenstephan’ is a German variety.

### 2.3. Dry Matter, Ash Content, Titratable Acidity, Pectin and Vitamin C

An assessment of the dry matter content of the elderflowers was performed by means of a gravimetric method using the standard AOAC method [16]. The fresh elderflower samples were precisely weighed (1.5 g) and dried at a temperature of 70 °C under a 3 kPa vacuum pressure until a constant weight was obtained. Measurements were performed in triplicate and expressed as percent (%).

Elderflower ash content was determined using the AOAC [16] 930.09 method. Measurements were performed in triplicate and expressed as percent (%). 

The titratable acidity of samples was determined by using a pH meter (IQ’s Scientific Instruments), according to Polish Norm [17]. The chopped samples were transferred to a volumetric flask (100 mL) which was then filled with water. Prepared samples were boiled and filtered after cooling down. About 10 mL of obtained filtrate was titrated with NaOH (0.1 mol/ dm^3^) up to pH 8.1. The measurements were performed in triplicate and expressed in grams (g) of malic acid per 100 g of dry matter (dm).

Pectin was determined by the method of Morris. Then, 50 mL of distilled water was added to 10 g of flowers. The mixture was boiled for 30 min and filtered afterwards. The obtained filtrate was transferred to a 50 mL volumetric flask and filled with distilled water until the investigations. Next, 10 mL of the solution was mixed with 50 mL of acetone and the sample was left for 1 h. It was then filtered through a dried and weighed filter and then dried at 75 °C for 4 h and weighed. Pectin content was calculated from the difference between the weight of the filter with the sediment after drying and the weight of the dried filter, considering the amount of pectin solution for determination and the sample weight. Results were provided in percent (%) [18].

Vitamin C was analyzed in accordance with Czaplicka [19]. The method consists in the oxidation of l-ascorbic acid to dehydroascorbic acid in an acid medium with a blue dye of 2,6-dichloroindophenol, followed by the reduction of the dye to the leuko (colorless) form, which takes on a red color at a pH of 4.2. 

The determination of ABTS and FRAP content was performed in methanol extracts (80% *v*/*v*, material to extracting agent ratio amounted to 1:5). The ABTS and FRAP antioxidant assays were determined as previously described by Oziembłowski et al. [20], respectively, using a Shimadzu UV-2401 PC spectrophotometer (Kyoto, Japan). All determinations were performed in triplicate. The results were expressed in μmol Trolox/100 g of dry matter (dm).

### 2.4. Determination of Sugar Content

The determination was made using the method described by Veberic et al. [21] using 87% acetonitrile. To clean the sample of impurities that could falsify the result, a system with a BAKER 12 G diaphragm vacuum pump was used. The sample prepared in this way was analyzed on a liquid chromatograph (HPLC) with a UV-VIS diode detector. Chromatographic analysis was carried out with an L-7455 liquid chromatograph (Merck-Hitachi, Tokyo, Japan) with an evaporative light scattering detector (PL-ELS 1000; Polymer Laboratories Ltd., Church Stretton, UK) and an L-7100 quaternary pump (Merck-Hitachi), equipped with a D-7000 HSM Multisolvent Delivery System (Merck-Hitachi), an L-7200 autosampler (Merck-Hitachi) and a Prevail Carbohydrate ES HPLC Column-W (250 × 4.6 mm, 5 μm; Alltech Inc., Nicholasville, KY, USA). Calibration curves (R^2^ = 0.9999) were created for glucose, fructose and sorbitol. All data were obtained in triplicate. Results are expressed as grams per 100 g of dry matter (dm) [18].

### 2.5. Analysis of Organic Acids by HPLC Method 

For the extraction of samples, the protocol described by Nawirska-Olszańska et al. [18] was followed. Organic acid analysis was carried out with a Dionex (Sunnyvale, CA, USA) liquid chromatographer equipped with an Ultimate 3000 LEDdetector, LPG-3400Apump, EWPS-3000SI autosampler, TCC-3000SD column thermostat, an AminexHPH-87H (300 mm × 7.8 mm) column with IGCationH (30 mm × 4.6 mm) Bio-Red precolumn and Chromeleon v.6.8 computer software.

Separation was conducted at 65 °C. The elution solvent was 0.001 N sulfuric acid. Samples (10 μL) were eluted isocratically at a flow rate of 0.6 mL^−1^. The compounds were monitored at 210 nm. All data were obtained in triplicate. The results were expressed as mg per 100 g of dry matter (dm).

### 2.6. Carotenoid Content UPLC-PDA-MS Analysis 

For the extraction and determination of carotenoids, a protocol described earlier was followed [18]. The powder samples of flowers (0.25 g) containing 10% MgCO_3_ were continuously shaken with 5 mL of hexane–acetone–methanol (2:1:1, *v*/*v*/*v*) containing 1% BHT, at 500 rpm (DOS-10L Digital Orbital Shaker, Elmi Ltd., Riga, Latvia) for 30 min in the dark. After the first extraction, the samples were centrifuged at 19,000× *g* for 10 min at 4 °C, and the supernatant was recovered. The samples were re-extracted and centrifuged under the same conditions. Supernatants were combined and evaporated to dryness. The pellet was re-extracted using 2 mL of 100% methanol, filtered through a hydrophilic PTFE 0.20 μm membrane (Millex Samplicity Filter, Merck) and used for analysis.

Compounds were separated with an ACQUITY UPLC BEH RP C18 column (1.7 μm, 2.1 mm × 100 mm, Waters Corp.) at 32 °C. The elution solvents were ACN:MeOH (7:3, *v*/*v*) (A) and 0.1% formic acid (B). Samples (10 μL) were eluted according to the linear gradient described previously [19]. Weak and strong needle solvents were ACN–MeOH (7:3, *v*/*v*) and 2-propanol, respectively.

The identification of carotenoids was carried out on the basis of fragmentation patterns and on the basis of PDA profiles. Where available, compounds were compared with authentic standards (their fragmentation pathways, retention times and PDA profiles). The runs were monitored at 450 and 427 nm. The PDA spectra were measured over the wavelength range of 200–800 nm in steps of 2 nm. Calibration curves were made from all-trans-β-carotene and all-trans-lutein. All incubations were carried out in triplicate. The results were expressed as milligrams per kilogram of dry matter (dm).

### 2.7. Identification and Quantification of Polyphenols Using the UPLC-PDA-MS Method 

Extraction Procedure: In order to prepare the stock solution, 1 ± 0.02 g of dried material was dissolved in 5 mL of 50% ethanol. The samples of plants were extracted using 50% ethanol via maceration in darkness at 4 °C. After 24 h, extracts were centrifuged at 3600 rpm for 20 min and the supernatants were collected and stored at −10 °C until testing. 

The extract for polyphenol analysis was prepared as described by Nawirska-Olszańska et al. [22]. An analysis of polyphenols was carried out using an ACQUITY Ultra Performance LC system (UPLC) equipped with binary solvent manager (Waters Corp., Milford, MA, USA), a UPLC BEH C18 column (1.7 μm, 2.1 mm 50 mm, Waters Corp.) and a Q-Tof Micro mass spectrometer (Waters, Manchester, UK) with an ESI source operating in negative and positive modes. The mobile phase consisted of aqueous 0.1% formic acid (A) and 100% acetonitrile (B). Samples (10 μL) were eluted according to the linear gradient described previously by Nawirska-Olszańska et al. [22]. The conditions of the mass spectrometer were a source block temperature of 130 °C, desolvation temperature of 350 °C, capillary voltage of 2.5 kV, cone voltage of 30 V and a desolvation gas (nitrogen) flow rate of 300 L/h. 

The compounds were monitored at 280 nm (flavan-3-ols and hydroquinones), 320 nm (phenolic acids), 340 nm (flavones) and 360 nm (flavonol glycosides). All experiments were carried out in triplicate. The results were corrected for dilution and expressed in mg/100 g of dry matter (dm).

### 2.8. Statistical Analysis

The obtained data were subjected to statistical analysis performed using Statistica v. 13.3 (StatSoft Polska, Kraków, Poland). Results were presented as means ± standard deviation (SD). Analysis of variance was also performed with ANOVA procedures. Significant differences (*p* ≤ 0.05) between mean values were determined by Duncan’s multiple-range test. A PCA analysis and a correlation between the chemical composition of the samples and the other tested parameters were also performed.

## 3. Results and Discussion

The tested samples of *Sambuci flos* were characterized by a high dry matter content, which ranged from 31.17% for ‘Sambo’ to 35.60% for ‘Weihenstephan’ (Table 1). In the study by Stefaniak et al. [23], the dry matter content was significantly lower (from 6.68% to 18.85%). Such significant differences may result primarily from the study of different varieties of elderberry flowers, as well as from weather conditions and harvest time.

The highest ash content was recorded in the ‘Weihenstephan’ variety (4.68%); this was almost twice as low in the ‘Sambo’ variety (2.41%). In the study by Stefaniak et al. (2019), [23] the ash content, similarly to the dry matter content, was significantly lower (from 0.533% to 1.564%).

The total acidity in the analyzed raw material ranged from 0.75 for ‘Sambo’ to 0.99 g/100 g for ‘Haschberg’. Using analysis of variance, it was shown that the varieties ‘Haschberg’ and ‘Albida’ had the highest acidity, and they differed statistically significantly from each other. However, the lowest result, statistically significantly different from the other values, was obtained for the ‘Samdal’ variety and wild elderberry (0.75 and 0.76 g/100 g). The values obtained for the remaining samples were similar.

The pectin content in elderberry flowers was low, ranging from 0.19 to 0.49% depending on the variety. The highest values were found for the varieties ‘Weihenstephan’ (0.49%) and ‘Haschberg’ (0.46%), and the lowest for ‘Bohatka’ (0.19%) and wild elderberry (0.20%). There are no reports in the literature regarding the pectin content in elderberry flowers.

The vitamin C content in the tested raw material ranged from 13.36 to 25.67 mg/100 g dm. Research conducted by Barros et al. [24], using the spectrophotometric method on freeze-dried flowers of *Sambucus nigra* L., showed that they contained 173 mg/100 g dm ascorbic acid. However, Socaci et al. [25] obtained a significantly higher result for fresh flowers—458.91 mg/100 g dm—and for dried flowers—141.88 mg/100 g dm. Such significant differences can be explained by the different methods used for determinations, as well as the atmospheric conditions and varieties used for testing.

It is known that the sugar content in the raw material affects its palatability. According to Barros et al. [24], elderberry flowers mainly contain three sugars: glucose, fructose and sucrose. It is also known that the dominant sugar in the raw material is glucose, while sucrose is present in the smallest amounts. Petruţ et al. [11] report, following other authors, that the sugar content in elderberry flowers is mainly glucose and fructose, depending on the variety (from 60% to 85%).

Elderflowers are usually used in preparing beverages, desserts and teas, where the sugar content affects their taste. Three sugars were identified in the tested elderberry flowers: glucose, fructose and sucrose (Table 2). Statistically, the ‘Sambo’ variety had the highest sugar content (glucose: 16.34 mg/100 g dm; fructose: 31.97 mg/100 g dm and sucrose 5.39 mg/100 g dm). None of the varieties clearly had the lowest content of all three sugars, but the wild form had the lowest sugar sum content. This was probably due to the lack of sucrose in these flowers. Moreover, it was found that the dominant saccharide was fructose, except for wild elderberry flowers, in which glucose dominated. Much lower sucrose contents were determined in all varieties compared to the results of other researchers. The analysis of elderberry flowers by Barros et al. [24] showed that they contained the most glucose (glucose: 32.3 mg/g; fructose: 26.6 mg/g; sucrose: 24.7 mg/g). In the study by Mikulic-Petkovsek et al. [26] conducted on flowers of various varieties and hybrids of elderberry, the content of total sugars was significantly higher in the *S. nigra* variety (71.44 g/kg dm). At the same time, in the cited studies, the highest sucrose level was 27.35 g/kg dm, contrary to other researchers.

The ratio of sugars to organic acids was the highest for the ‘Sambo’ variety (1.56), which is the sweetest variety in terms of organoleptic sensation, while the flowers of the ‘Haschberg’ variety are the most sour. Such differentiation is confirmed by the research of Mikulic-Petkovsek et al. [26].

Five organic acids were identified in elderberry flowers: citric acid, malic acid, fumaric acid, tartaric acid and shikimic acid, as shown in Table 3. The most acids were determined in the ‘Haschberg’ variety (mg/100 g dm). The ‘Albida’ variety was unique because it had the highest amount of malic acid (43.02 mg/100 g dm) and the least amount of other acids, but the sum of all acids placed it in second place in terms of the content of organic acids (52.59 mg/100 g dm). The variety ‘Samdal’ (33.14 mg/100 g dm) and the wild form (33.32 mg/100 g dm) had the lowest contents of organic acids. These results are consistent with those obtained by Mikulic-Petkovsek et al. [26].

Malic and citric acids were most abundant and, combined, represented from 75% to 90% of the total analyzed organic acids (Table 3). In all tested varieties, the lowest levels of fumaric acid were determined (from 0.14 to 0.47 mg/100 g dm) and this was similar to results obtained by Mikulic-Petkovsek et al. [26]. In these studies, the content of fumaric acid was also the lowest (0.15 to 0.92 g/kg dm) and for wild elderberry it was determined as 0.43 g/kg dm

Three carotenoids were identified in the tested varieties of elderberry flowers: β-carotene, lutein, lycopene and trace amounts of the chlorophyll pigment, pheophytin A (Table 4). Based on the analysis of variance, it was noticed that the variety with the statistically significantly highest content of the tested compounds was ‘Samdal’ (respectively, 16.05; 4.44; 1.19; 0.51 μg/g dm). It was estimated that the content of β-carotene in this sample accounted for over 72% of the composition of all carotenoids detected in it. The remaining ingredients were lutein (20%), lycopene (5.4%) and pheophytin A (2.3%). The flowers with the lowest contents of these natural pigments were ‘Albida’ (respectively, 3.09; 1.02; 0.15 μg/g dm). This variety differed significantly from the other inflorescences. Moreover, similarly to the ‘Haschberg’ sample and the wild form, the presence of pheophytin A was not identified in it. The percentage composition of individual compounds in the ‘Albida’ variety was β-carotene—72.5%, lutein—23.9% and lycopene—3.5%.

According to the experiment conducted by Barros et al. [24], it was shown that *Sambuci flos* contained 18.4 μg/g dm of β-carotene and 5.34 μg/g dm of lycopene. These values were higher than those obtained in our research.

The tested *Sambuci flos* did not have a high amount of natural pigments; the lycopene content was especially low. In the tested varieties of elderberry flowers, the presence of pheophytin A, i.e., chlorophyll pigment (olive green), was also detected, which is characteristic of green parts of plants and is produced, for example, under the influence of acids.

A total of 41 polyphenolic compounds were identified in the seven variants of elderberry flowers examined (Appendix A). The graph in Appendix A shows an example of a chromatogram for wild elderflowers. The greatest diversity of identified polyphenolic compounds was found in flowers from the ‘Bohatka’ variety (34), and the lowest in flowers from the ‘Albida’ variety (21). The same number of polyphenolic compounds (28) were identified in the ‘Samdal’ and ‘Sambo’ varieties (Table 5). The compounds identified in the flowers of all varieties were Quinic acid, 3-Caffeoylquinic acid, 5-Caffeoylquinic acid, 4-Caffeoylquinic acid, Quercetin hexoside pentoside, 5-p-Coumaroylquinic acid, Quercetin dihexoside, 3-Feruloyl-quinic acid, Quercetin-3 -rutinoside, Kaempferol-3-rutinoside and Dicaffeoylquinic acid dimer. Protocatechuic acid was identified in flowers from the ‘Haschberg’ variety and was not identified in any other flowers tested. The flowers obtained from the wild form contained as many as three compounds that were not present in the other flowers: cis-3-Caffeoylquinic acid, cis-4-Caffeoylquinic acid and Isorhamnetin acetyl hexoside. The obtained results show that the cis form of Caffeoylquinic acid occurred only in wild flowers, while 3-Caffeoylquinic acid and 5-Caffeoylquinic acid were present in all tested forms. The reasons for this can be found in the place of obtaining the raw material and the influence of climatic conditions during the development of inflorescences. B-type procyanidin dimer occurred in all varieties except the wild form; in this case, the reasons may be analogous to the occurrence of certain compounds only in the flowers of the wild form. The majority of polyphenolic compounds were identified in the flowers of the ‘Bohatka’ variety, but no compound specific to these flowers was identified. In the wild form, two fewer compounds were identified than in ‘Bohatka’ but, here, as many as three compounds not found in other flowers were determined.

In a review paper by Młynarczyk et al. [13], 43 polyphenolic compounds were identified, which is 2 compounds more than in the presented research, but not all compounds overlapped. This is certainly due to the different varieties tested, as well as the cultivation and climatic conditions. Small amounts of quinic acid, which has not been identified by other researchers, were identified in the examined flowers. Only in the flowers of the ‘Haschberg’ variety were traces of protocatechuic acid identified, which is a metabolite of cyanidin-glucosides that has not been identified by other researchers. 

It was found that the tested raw material was a rich source of polyphenolic substances, the content of which ranged from 197.18 to 394.26 mg/100 g dm. The values obtained for individual research objects (varieties) were statistically different, while only flowers from the ‘Samdal’ and ‘Sambo’ varieties constituted one homogeneous group. Among the identified polyphenols present in all tested variants, the highest concentrations were of Quercetin-3-rutinoside, from 36.58 mg/100 g dm in the flowers of the ‘Haschberg’ variety to 71.32 mg/100 g dm in the flowers of the ‘Samdal’ variety. This has been confirmed by other researchers, as presented in the review by Młynarczyk et al. [13]. The content of Quercetin-3-rutinoside in various papers ranged from 11.6 to 42.3 mg/g dm, which was slightly higher than in the tested varieties. Another compound that was determined in larger amounts (53.96–92.65 mg/100 g dm) was Caffeoylquinic acid dimer, but it was not determined in the ‘Sambo’ and ‘Samdal’ varieties. According to data from the paper of Młynarczyk et al. [13], Caffeoylquinic acid dimer was not identified by other researchers, but they identified other derivatives of this acid.

‘Albida’ flowers turned out to be a statistically significantly different variety from other samples due to their having the lowest amount of polyphenolic compounds in their composition (197.18 mg/100 g dm).

In the research presented by Młynarczyk et al. [27] on various varieties of fresh *Sambuci flos*, three of which were wild-growing forms and the remaining varieties, ‘Haschberg’, ‘Sampo’ and ‘Samyl’ obtained from a plantation in Poznań, the content of total polyphenols was determined at a level that was within in the range of 55.31–81.56 mg/g dm.

The research carried out on methanolic extracts of elderberry flowers and the analysis of variance in a one-factor way allowed for the conclusion that, statistically, the highest antioxidant activity (Table 6.) in both methods was observed in the ‘Weihenstephan’ variety (ABTS: 449.9; FRAP: 348.9 μMol Trolox/g dm). However, the ‘Albida’ variety, in two tests, also turned out to have the significantly lowest antioxidant capacity (ABTS: 249.5; FRAP: 199.8 μMol Trolox/g dm). It was also noticed that the ‘Samdal’ and ‘Haschberg’ varieties differed significantly from the ‘Sambo’ flowers and the wild form. For ‘Bohatka’ flowers, in both cases there were statistically significant differences compared to the other samples. Moreover, it turned out that the reducing power of FRAP showed a strong positive correlation (r = 0.93) with the content of total polyphenols in the analyzed *Sambuci flos*. For antioxidant activity measured by the ABTS method, a strong positive correlation was also obtained with the contents of the above-mentioned ingredients (r = 0.95).

Elderberry flowers were examined by Młynarczyk et al. [27] in terms of antioxidant capacity measured by a method based on the measurement of the ability to scavenge the ABTS cation radical. For methanol extracts from fresh flowers from plantations, comparable values were obtained, which were in the range of approximately 400–450 μMol Trolox/g dm. The results for the dried raw material of the same origin were lower at approx. 300–350 μMol Trolox/g dm. However, dried *Sambuci flos*, obtained from natural places of occurrence, were characterized by antioxidant activity at levels of approx. 230–300 μMol Trolox/g dm; for fresh inflorescences, this was slightly higher at approx. 300–350 μMol Trolox/g dm. In the experiment of Wołosiak et al. [28], conducted on methanol extracts of dried elderberry flowers collected away from busy public roads, 239 μMol of Trolox/g dm was obtained. The results of antioxidant activity tests presented as ABTS by Stuper-Szablewska et al. [1] (50.28 μMol Trolox/g dm) differ significantly from the results obtained in this work and by other researchers. The plant metabolism and the synthesis of secondary metabolites are the results of several intrinsic factors, such as genetic factors that characterize the cultivars, as well as the season of flowering.

Based on the results obtained, a PCA analysis was also performed. The input data referred to seven cases (varieties) and 23 variables as average values from Table 1, Table 2, Table 3, Table 4 and Table 6 and the sum of phenolic compounds in Table 5. PCA analysis showed that the first two PCA factors (i.e., PCA 1 and PCA 2) explain 57.28% of the total variance and the first three PCA factors explain 77.44%. The eigenvalues were 7.331, 5.844 and 4.636 for PCA 1, PCA 2 and PCA 3, respectively. Due to the similarity of the PCA 1-PCA 2 and PCA 1-PCA 3 graphs, we decided to only present the figures for the PCA1-PCA2 pair.

An analysis of Figure 1 indicates the existence of two compact clusters of points representing the analyzed varieties. The first group consists of the ‘Samdal’, ‘Sambo’, ‘Weihenstephan’ and ‘Bohatka’ varieties, and the second group consists of the ‘Albida’ and ‘Haschberg’ varieties. Outside these two groups was wild elderberry, which differed the most from the other two groups in terms of the chemical composition of its flowers.

When interpreting plots of factor coordinates of variables (Figure 2), attention should be paid to the length of the vectors and the angle between them. The longer the vector, the greater the share of a given feature (or variable, here chemical compounds) in the values of the two analyzed components (here PCA1 and PCA2). In turn, the smaller the angle between the vectors, the more similar the impact of these features (variables) on the analyzed components, i.e., the closer together the vectors (points) are located, the greater the positive correlation between the variables. The larger the angle the vectors form, the less correlated the variables are. If the vectors are perpendicular, then the variables are uncorrelated. If they form an angle close to 180°, they are negatively correlated.

There are many relationships between the studied variables, but an analysis of Figure 2 allows us to identify several of the most important relationships. The following variables had the greatest influence on the PCA model: malic acid, sum of acids, β-carotene, lycopene and S/A ratio. The following variables had the smallest impact on the PCA model: glucose and dry matter. Individual groups of variables were highly correlated, including ABTS, FRAP, pheophytin A, fumaric acid, lycopene and S/A ratio or another group of variables such as saccharose, fructose, sum of sugars and tartaric acid. Negatively correlated with each other were, among others, such variables as β-carotene and sum of acids. No correlation was found between many pairs of variables, including the following pairs: malic acid–sum of sugars or β-carotene–fructose. The variables and their relationships shown in Figure 2 make it easier to predict individual correlations relating to the raw material, which is elderberry flowers of various varieties.

## 4. Conclusions

This research confirms the assumption that elderberry flowers are rich in bioactive ingredients and, at the same time, it has been proven that their content depends on the flower variety. The wild form turned out to be the richest in bioactive ingredients, while among the varietal plants it was ‘Weihenstephan’. Both of these forms were characterized by a high content of polyphenols and antioxidant activity. However, flowers from the ‘Bohatka’ variety had the most diverse composition of polyphenol compounds, while the flowers of the white fruits ‘Albida’ variety had the least diverse composition. The ‘Sambo’ variety contained the most sugars, and this variety also had the highest ratio of sugars to organic acids, i.e., it was characterized by the highest perceived sweetness. The ‘Bohatka’ variety was also characterized by the highest content of vitamin C. The content of carotenoids varied greatly. The largest number of compounds from this group was determined in flowers from the ‘Samdal’ and ‘Sambo’ cultivars. The flowers of these two varieties also had similar polyphenol contents. The PCA analysis showed general differences in the chemical composition of elderberry flowers, depending on the variety, in relation to 23 variables; it also showed their mutual correlations and the strength of their influence on the PCA model. Based on the research conducted, it is difficult to clearly select the best variety. Depending on the processing needs, the variety that contains the most ingredients that the processor cares about should be chosen.

## Figures and Tables

**Figure 1 foods-13-02560-f001:**
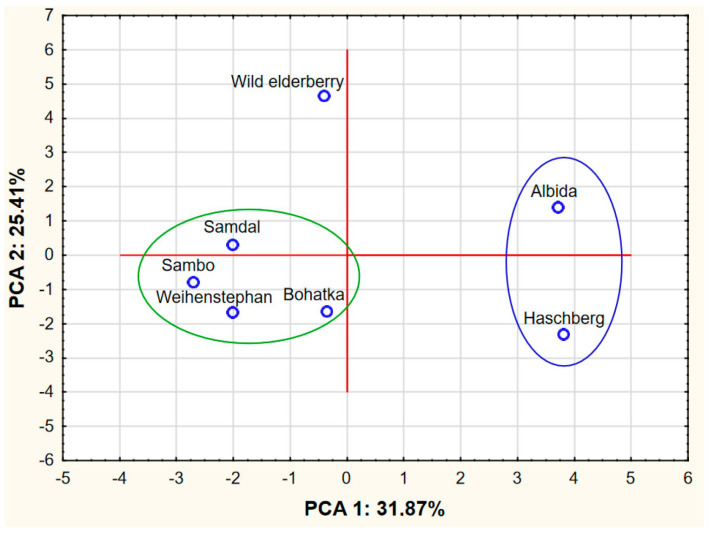
Chart of factor coordinates of cases (varieties) in the PCA model.

**Figure 2 foods-13-02560-f002:**
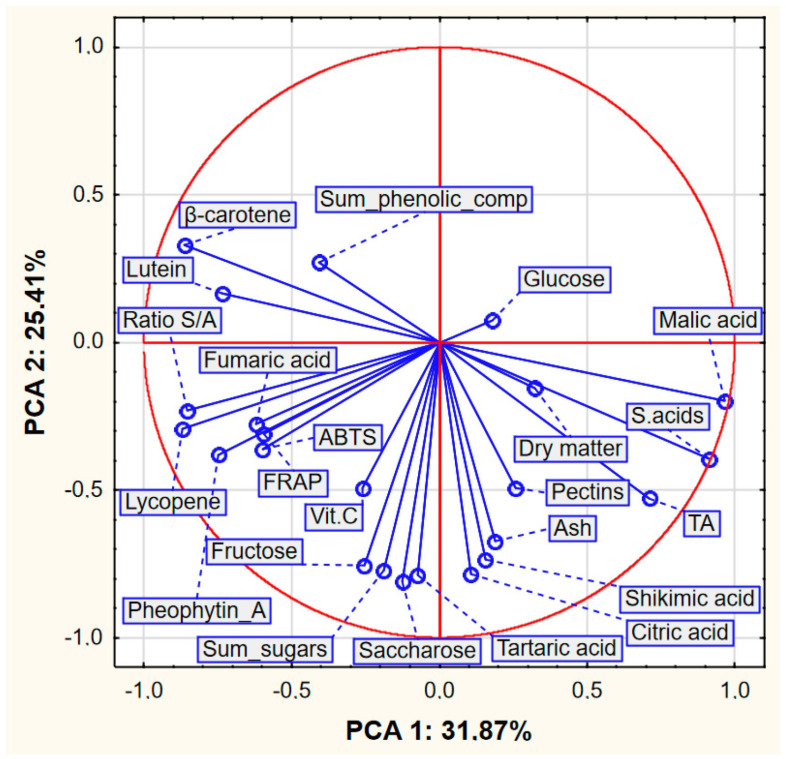
Chart of factor coordinates of variables in the PCA model.

**Table 1 foods-13-02560-t001:** Content of dry matter, ash, total acidity, pectin and vitamin C in elderberry flowers of various varieties.

Variety	Dry Matter [%]	Ash [%]	Titratable Acidity [g/100 g]	Pectin [%]	Vitamin C [mg/100 g]
Albida	34.19 ± 2.81 b	3.42 ± 0.67 c	0.95 ± 0.08 b	0.42 ± 0.04 c	13.36 ± 0.58 g
Bohatka	33.36 ± 2.67 c	3.88 ± 0.38 b	0.84 ± 0.03 e	0.19 ± 0.01 f	25.67 ± 1.56 a
Haschberg	34.25 ± 2.44 b	3.94 ± 0.43 b	0.99 ± 0.09 a	0.46 ± 0.03 b	21.15 ± 1.29 c
Sambo	31.17 ± 2.93 d	2.45 ± 0.85 e	0.86 ± 0.06 d	0.40 ± 0.03 d	16.61 ± 0.93 f
Samdal	33.79 ± 2.27 c	3.31 ± 0.78 c,d	0.75 ± 0.04 f	0.28 ± 0.01 e	19.52 ± 0.56 d
Weihenstephan	35.60 ± 2.28 a	4.68 ± 0.56 a	0.89 ± 0.07 c	0.49 ± 0.03 a	22.59 ± 0.99 b
Wild elderberry	33.30 ± 2.35 c	2.41 ± 0.24 e	0.76 ± 0.08 f	0.20 ± 0.01 f	18.77 ± 1.22 e

Values are means ± standard deviation, n = 3; in columns mean values with different letters are significantly different at *p* < 0.05.

**Table 2 foods-13-02560-t002:** Sugar content in different varieties of elderberry flowers.

Variety	Fructose	Saccharose	Glucose	Sum	Ratio S/A
	g/100 g dm				
Albida	27.65 ± 1.65 d	3.93 ± 0.08 c	13.11 ± 0.08 d,e	44.71	0.85
Bohatka	30.58 ± 2.54 b	5.47 ± 0.78 a	15.05 ± 0.08 c	51.10	1.12
Haschberg	25.07 ± 1.76 e	3.86 ± 0.07 c	16.47 ± 0.12 a	45.41	0.75
Sambo	31.97 ± 2.98 a	5.39 ± 0.63 a	16.34 ± 0.15 a	53.70	1.56
Samdal	30.69 ± 2.83 b	4.29 ± 0.08 b	13.60 ± 0.09 d	48.59	1.47
Weihenstephan	29.25 ± 1.89 c	3.85 ± 0.09 c	10.83 ± 0.06 f	43.95	1.20
Wild elderberry	13.52 ± 1.28 f	0.00	15.63 ± 0.11 a,b	29.15	0.87

Values are means ± standard deviation, n = 3; in columns mean values with different letters are significantly different at *p* < 0.05.

**Table 3 foods-13-02560-t003:** Organic acid composition of different varieties of elderberry (mg/100 g dm).

Variety	Citric Acid	Malic Acid	Fumaric Acid	Tartaric Acid	Shikimic Acid	Sum
	mg/100 g dm					
Albida	5.27 ± 0.41 e	43.02 ± 2.71 a	0.14 ± 0.01 d	2.48 ± 0.01 e	1.68 ± 0.41 e	52.59
Bohatka	8.15 ± 0.33 b	31.38 ± 3.96 b	0.31 ± 0.01 b	3.16 ± 0.31 d	2.79 ± 0.33 b	45.79
Haschberg	9.57 ± 0.23 a	42.69 ± 1.21 a	0.29 ± 0.01 c	4.52 ± 0.23 a	3.23 ± 0.23 a	60.30
Sambo	7.24 ± 0.09 c	21.10 ± 1.73 d	0.28 ± 0.02 c	3.45 ± 0.09 c	2.41 ± 0.09 c	34.48
Samdal	7.51 ± 0.09 c	19.41 ± 1.97 e	0.32 ± 0.01 b	3.51 ± 0.09 c	2.39 ± 0.09 c	33.14
Weihenstephan	6.64 ± 0.07 d	23.11 ± 1.21 c	0.47 ± 0.01 a	4.33 ± 0.07 b	1.99 ± 0.01 d	36.54
wild elderberry	5.33 ± 0.21 e	23.42 ± 1.19 c	0.32 ± 0.01 b	2.53 ± 0.01 e	1.72 ± 0.01 e	33.32

Values are means ± standard deviation, n = 3; in columns mean values with different letters are significantly different at *p* < 0.05.

**Table 4 foods-13-02560-t004:** Carotenoid content in elderberry flowers.

Variety	β-Carotene [μg/g dm]	Lutein [μg/g dm]	Lycopene [μg/g dm]	Pheophytin A [μg/g dm]
Albida	9.34 ± 0.83 d	1.02 ± 0.08 f	0.15 ± 0.01 g	0
Bohatka	9.95 ± 0.74 d	1.64 ± 0.09 e	1.02 ± 0.09 c,d	0.55 ± 0.06 c
Haschberg	3.09 ± 0.26 e	1.72 ± 0.16 e	0.21 ± 0.07 f	0
Sambo	14.58 ± 1.16 b	3.15 ± 0.31 b	2.45 ± 1.52 a	1.61 ± 0.76 a
Samdal	16.05 ± 0.52 a	4.44 ± 0.26 a	1.19 ± 0.09 c	0.51 ± 0.05 c
Weihenstephan	12.90 ± 0.56 c	2.37 ± 0.21 d	1.67 ± 0.98 b	0.75 ± 0.09 b
Wild elderberry	12.54 ± 0.98 c	2.78 ± 0.13 c	0.59 ± 0.09 e	0

Values are means ± standard deviation, n = 3; in columns mean values with different letters are significantly different at *p* < 0.05.

**Table 5 foods-13-02560-t005:** Content of phenolic compounds (mg/100 g dm) in elderberry flowers.

Compound	Albida	Bohatka	Haschberg	Sambo	Samdal	Weihenstephan	Wild Elderberry
Quinic acid	0.01 ± 0.00 c	0.02 ± 0.00 b	0.01 ± 0.00 c	0.02 ± 0.00 b	0.02 ± 0.00 b	0.02 ± 0.00 b	0.03 ± 0.00 a
Protocatechuic acid	nd	nd	0.01 ± 0.00 a	nd	nd	nd	nd
Hydroxybenzoic acid	25.07 ± 3.12 e	36.76 ± 2.22 a	29.17 ± 3.11 d	35.07 ± 2.32 b	31.26 ± 1.21 c	34.16 ± 2.19 b	nd
Caffeoyl N-tryptophan	nd	0.02 ± 0.00 a	0.02 ± 0.00 a	nd	nd	nd	nd
Caffeoylhexose	2.98 ± 0.01 d	4.55 ± 0.01 d	nd	6.49 ± 0.04 a	6.22 ± 0.04 b	3.86 ± 0.02 c	nd
3-Caffeoylquinic acid	2.39 ± 0.01 d	3.98 ± 0.02 c	5.22 ± 0.14 b	2.99 ± 0.08 d	3.02 ± 0.09 d	3.86 ± 0.12 c	10.01 ± 0.15 a
Caffeoylquinic acid dimer	53.96 ± 4.33 d	89.56 ± 5.28 a	65.32 ± 3.48 c	nd	nd	78.99 ± 4.17 b	12.65 ± 1.42 e
cis-3-Caffeoylquinic acid	nd	nd	nd	nd	nd	nd	92.65 ± 6.25 a
5-Caffeoylquinic acid	4.12 ± 0.08 e	8.76 ± 0.09 d	14.33 ± 0.89 c	10.98 ± 0.16 c	11.24 ± 0.18 c	15.76 ± 0.65 b	25.48 ± 1.64 a
cis-4-Caffeoylquinic acid	nd	nd	nd	nd	nd	nd	13.88 ± 0.95 a
B-type procyanidin dimer	9.28 ± 0.24 d	15.72 ± 0.99 a	11.34 ± 0.97 c	13.53 ± 0.89 b	13.97 ± 0.88 b	16.51 ± 1.09 a	nd
4-Caffeoylquinic acid	0.92 ± 0.02 c,d	0.99 ± 0.07 c	0.97 ± 0.03 c	1.78 ± 0.09 a	1.86 ± 0.09 a	0.98 ± 0.04 c	1.21 ± 0.08 b
Quercetin hexoside pentoside	1.23 ± 0.09 c	4.87 ± 0.32 b	9.76 ± 0.97 a	10.12 ± 1.06 a	10.22 ± 1.07 a	3.55 ± 0.23 b	9.81 ± 0.98 a
Kaempferol dihexoside	43.76 ± 2.09 c	nd	nd	59.76 ± 3.66 a	60.14 ± 3.66 a	61.54 ± 4.78 a	30.38 ± 2.11 d
5-p-Coumaroylquinic acid	2.67 ± 0.21 e	11.8 ± 0.99 a	7.98 ± 0.69 c,d	6.21 ± 0.33 d	6.24 ± 0.22 d	8.33 ± 0.82 b,c	3.98 ± 0.03 e
Quercetin dihexoside	0.99 ± 0.04 e	2.01 ± 0.22 b	1.98 ± 0.16 b	1.23 ± 0.11 d	1.33 ± 0.08 c.d	1.49 ± 0.09 c	2.27 ± 0.29 a
3-Feruloyl-quinic acid	1.99 ± 0.04 e,f	2.54 ± 0.19 c	2.63 ± 0.22 b	2.49 ± 0.18 d	2.46 ± 0.12 d	2.11 ± 0.09 e	2.93 ± 0.29 a
4-Feruloyl-quinic acid	nd	2.98 ± 0.02 c	3.52 ± 0.03 b	nd	nd	nd	9.53 ± 0.56 a
Quercetin-3-O-rhamnosyl hexoside	nd	4.76 ± 0.36 c	6.58 ± 0.39 b	nd	nd	nd	8.66 ± 0.84 a
Quercetin dihexoside 1	0.99 ± 0.01 c	nd	1.12 ± 0.09 b	1.30 ± 0.09 a	1.28 ± 0.08 a	nd	1.32 ± 0.08 a
Ferulic acid hexoside	nd	15.41 ± 1.25 b	30.98 ± 2.33 a	nd	nd	16.29 ± 4.66 b	nd
Quercetin dihexoside 2	nd	0.82 ± 0.06 c	0.86 ± 0.05 c	1.02 ± 0.09 b	0.98 ± 0.08 b	nd	1.37 ± 0.09 a
Isorhamnetin dihexoside	36.54 ± 2.67 d	36.66 ± 1.92 d	33.58 ± 2.26 d,e	66.44 ± 5.43 b	69.32 ± 6.94 b	50.98 ± 4.29 c	99.87 ± 3.22 a
Quercetin acetyldihexoside	nd	2.43 ± 0.08 d	2.99 ± 0.17 c	3.17 ± 0.27 b	3.15 ± 0.21 b	3.24 ± 0.09 a	nd
Quercetin-3-rutinoside	nd	13.22 ± 0.13 c	nd	nd	nd	13.42 ± 0.29 b	13.65 ± 0.25 a
Quercetin-3-O-rhamnosyl hexoside	0.28 ± 0.02 f	0.95 ± 0.08 e	1.23 ± 0.15 d	1.67 ± 0.22 c	nd	2.44 ± 0.22 b	2.72 ± 0.27 a
Quercetin-3-O-rhamnosyl hexoside	nd	nd	3.65 ± 0.22 b	3.42 ± 0.32 b	3.58 ± 0.29 b	6.98 ± 0.53 a	3.22 ± 0.26 b,c
Kaempferol-3-rutinoside	5.62 ± 0.04 e	9.29 ± 0.09 d	11.27 ± 0.99 a	9.74 ± 0.86 b	9.44 ± 0.74 c	10.98 ± 1.06 a,b	13.83 ± 1.25 a
Isorhamnetin-3-rutinoside	nd	0.88 ± 0.06 a	nd	0.68 ± 0.01 b	0.54 ± 0.01 c	nd	nd
Dicaffeoylquinic acid 1	nd	8.64 ± 0.63 b	nd	nd	nd	nd	14.37 ± 0.31 a
Dicaffeoylquinic acid dimer	0.10 ± 0.00 e	0.29 ± 0.01 a	0.21 ± 0.01 c	0.19 ± 0.01 c	0.18 ± 0.01 d	0.24 ± 0.01 b	0.25 ± 0.02 b
Isorhamnetin hexoside	nd	6.87 ± 0.33 a	nd	nd	nd	nd	nd
Isorhamnetin acetyl hexoside pentoside	nd	1.99 ± 0.09 d	2.21 ± 0.19 c	1.67 ± 0.09 e	1.34 ± 0.09 f	2.96 ± 0.22 b	3.74 ± 0.29 a
Dicaffeoylquinic acid 2	nd	0.01 ± 0.00 c	0.03 ± 0.00 c	0.11 ± 0.01 a	0.12 ± 0.02 a	0.07 ± 0.00 b	0.09 ± 0.00 b
Dicaffeoylquinic acid 3	nd	0.02 ± 0.00 c	0.01 ± 0.00 c	0.09 ± 0.00 a	0.08 ± 0.00 a	0.02 ± 0.00 c	0.05 ± 0.00 b
Isorhamnetin acetyl hexoside	nd	0.46 ± 0.01 b	0.53 ± 0.01 a	0.33 ± 0.01 c	0.35 ± 0.01 c	0.61 ± 0.01 a	nd
Coumaroylquinic acid derivative	nd	nd	nd	nd	nd	nd	1.22 ± 0.61 a
3-O-caffeoyl-4-O-p-coumaroylquinic acid	1.25 ± 0.09 f	5.96 ± 0.43 b	4.39 ± 0.26 d	5.27 ± 0.33 c	5.32 ± 0.23 c	3.87 ± 0.21 e	6.44 ± 0.44 a
Dihydrokaempferol-O-hexoside	3.01 ± 0.00 f	3.32 ± 0.00 d	3.46 ± 0.00 c	3.63 ± 0.00 a	3.22 ± 0.00 e	3.52 ± 0.00 b	3.34 ± 0.00 d
Naringenin derivative	0.02 ± 0.00 d	0.04 ± 0.00 c	nd	0.09 ± 0.00 a	0.07 ± 0.00 b	nd	nd
Naringenin hexoside	nd	0.03 ± 0.00 d	4.03 ± 0.00 a	0.05 ± 0.00 c	0.09 ± 0.00 b	0.06 ± 0.00 c	4.41 ± 0.00 a
sum	197.18 f	296.61 c	259.49 e	249.54 d	246.04 d	377.76 b	394.26 a

nd—not detected. Values are means ± standard deviation, n = 3; in columns mean values with different letters are significantly different at *p* < 0.05.

**Table 6 foods-13-02560-t006:** Antioxidant activity of elderberry flowers.

Variety	ABTS [μMol Trolox/g dm]	FRAP [μMol Trolox/g dm]
Albida	294.5 ± 11.53 e	199.8 ± 11.92 e
Bohatka	421.0 ± 13.23 b	321.0 ± 22.91 b
Haschberg	344.4 ± 16.64 d	238.3 ± 22.33 d
Sambo	378.9 ± 15.16 c	276.5 ± 21.53 c
Samdal	335.0 ± 17.88 d	232.5 ± 19.98 d
Weihenstephan	449.9 ± 22.98 a	348.9 ± 21.44 a
Wild elderberry	371.1 ± 17.22 c	275.4 ± 14.42 c

Values are means ± standard deviation, n = 3; in columns mean values with different letters are significantly different at *p* < 0.05.

## Data Availability

The original contributions presented in the study are included in the article/Appendix A, further inquiries can be directed to the corresponding author.

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
