# Peer review of "Elderflowers (Sambuci flos L.): A Potential Source of Health-Promoting Components"

_foods, 2024, doi:10.3390/foods13162560_

Round 1

Reviewer 1 Report

Comments and Suggestions for Authors

The current paper address an interesting subject. 

The chemical composition and bioactive compounds of sambucus co products is a field that deserves consideration especially since the beginning of the covid era, where it was once again discovered that sambucus flowers and fruits extracts are very effective in treatment of flu and lung disease prevention. 

The authors should consider the following observations:

1. Introduction section is very long, it is suggested to have one page, and to provide a comprehensive background and a hypothesis of the study. 

2. Material and methods chapter is clear and well done written. 

3. The results chapter is missing. 

The authors should consider presenting the results in a separate chapter and after to discuss the results in the Discussion section. 

Please follow the instructions for authors as indicated on the journal website. 

Presented in this way, the results are redundant and the information is repeated. Please remove the values when presenting the results are they are already presented in the mentioned table. 

Was the mineral composition determined? The soil usually has a significant effect on these nutrients. 

Also, what about the toxic compounds? Elderly co products are known for their high content of sambunigrin and niagerin which are toxic if ingested in higher amounts. 

The results presented in Table 5 are very interesting. However the authors should  present the name of each polyphenol instead of numbers. Also, they can be categorized in acids, flavonoids and so on. 

Eventually place the table in landscape for better data visualisation.  

In the PCA analysis,  Figure 1 and 2, is covers 57 of the variability. Have the authors also explored the PC1 with PC3? How much covers PC3? How much is the eigenvalue?

Good luck with your work. 

Author Response

Comments 1. Introduction section is very long, it is suggested to have one page, and to provide a comprehensive background and a hypothesis of the study. 

Response 1. The text has been shortened and research hypothesis has been added.

Comments 2. Material and methods chapter is clear and well done written. 

Response 2. Thank you!

Comments 3, 4, 5. The results chapter is missing. The authors should consider presenting the results in a separate chapter and after to discuss the results in the Discussion section. Please follow the instructions for authors as indicated on the journal website. 

Response 3, 4, 5. In our case, the results were discussed together in „Results and discussion” chapter, which is permitted by the journal's editorial office.

Comments 6. Presented in this way, the results are redundant and the information is repeated. Please remove the values when presenting the results are they are already presented in the mentioned table. 

Response 6. The results are presented in the text to make them more readable in discussions with other authors.

Comments 7. Was the mineral composition determined? The soil usually has a significant effect on these nutrients. 

Response 7. The soil composition was not tested. It does have an impact on the development and chemical composition of plants, but in the case of shrubs and trees it is of marginal importance, so the soil composition was not tested.

Comments 8. Also, what about the toxic compounds? Elderly co products are known for their high content of sambunigrin and niagerin which are toxic if ingested in higher amounts. 

Response 8. Since the flowers were dried, no toxic compounds were determined in them. Sambunigrin and niagerin are significantly decomposed by temperature and the drying process.

Comments 9. The results presented in Table 5 are very interesting. However the authors should  present the name of each polyphenol instead of numbers. Also, they can be categorized in acids, flavonoids and so on. 

Response 9. Of course, they can be classified as acids, flavanoids, etc., but we decided to arrange them according to the times of appearance, and they do not coincide with the division into individual fractions.

Comments 10. Eventually place the table in landscape for better data visualisation.  

Response 10. We ultimately decided to leave the table in its current layout.

Comments 11. In the PCA analysis,  Figure 1 and 2, is covers 57 of the variability. Have the authors also explored the PC1 with PC3? How much covers PC3? How much is the eigenvalue?

Response 11. PCA 3 explains 22.16% of the total variance. The eigenvalues were 7.331, 5.844 and 4.636 for PCA 1, PCA 2 and PCA 3, respectively. Due to the similarity of the PCA 1-PCA 2 and PCA 1-PCA 3 graphs, it was decided to present the diagrams only for the PCA1-PCA2 pair. That information was included in our paper.

Reviewer 2 Report

Comments and Suggestions for Authors

This manuscript analyzed the different chemical components and antioxidant capacity of six different cultivated and wild Elderflower varieties. However, the overall experimental design of the manuscript is too simple and does not involve the mechanism analysis that causes differences between different varieties and between wild and cultivated species. In addition, the presentation of data in the manuscript is too simplistic, resulting in insufficient logical coherence between the results and the discussion. Based on this, we believe that the paper needs major revisions.

Major Issues:

1. The abstract should clearly indicate which varieties are better;

2. The preface section is too verbose and lacks emphasis on key points, and should be streamlined and refined;

3. Insufficient summary of current research progress on the composition and health of Elderflower;

4. Is it reasonable to use fruits and flowers for some indicators in the method?

5. The results and discussion sections should be written separately, rather than a single headline and all of them should be completed.

Additional Minor Issues:

1. Discussion of the results OR of the Results and Discussion;

2. Line 266: 0.533%÷1.564% ?

3. p should be italicized;

4. The reference format is not consistent.

Comments on the Quality of English Language

no

Author Response

Major Issues:

Comments 1, 2, 3. The abstract should clearly indicate which varieties are better. The preface section is too verbose and lacks emphasis on key points, and should be streamlined and refined. Insufficient summary of current research progress on the composition and health of Elderflower.

Response 1, 2, 3. It is written in abstract that the best variety is Weihenstephan, but similar results were obtained for the wild variety of black elderberry. The preface has been corrected and shortened.

Comments 4. Is it reasonable to use fruits and flowers for some indicators in the method?

Response 4. This was an error in the description of the methodology and has been corrected.

Comments 5. The results and discussion sections should be written separately, rather than a single headline and all of them should be completed.

Response 5. In our case, the results were discussed together in „Results and discussion” chapter, which is permitted by the journal's editorial office.

Additional Minor Issues:

Additional comments 1. Discussion of the results OR of the Results and Discussion.

Additional response 1. It was revised for „Results and Discussion”.

Additional comments 2. Line 266: 0.533%÷1.564% ?

Additional response 2. These values were given by the authors of the paper Stefaniak et al. 2019

Additional comments 3. p should be italicized.

Additional response 3. It has been corrected.

Additional comments 4. The reference format is not consistent.

Additional response 4. The reference has been revised and complies with the Foods requirements.

Round 2

Reviewer 2 Report

Comments and Suggestions for Authors

The author has made the required modifications and it is recommended to accept them

Comments on the Quality of English Language

The author has made the required modifications and it is recommended to accept them